# Stable Consistency Tuning: Understanding and Improving Consistency Models

## Abstract

Diffusion models achieve superior generation quality but suffer from slow generation speed due to the iterative nature of denoising. In contrast, consistency models, a new generative family, achieve competitive performance with significantly faster sampling. These models are trained either through consistency distillation, which leverages pretrained diffusion models, or consistency training/tuning directly from raw data. In this work, we propose a novel framework for understanding consistency models by modeling the denoising process of the diffusion model as a Markov Decision Process (MDP) and framing consistency model training as the value estimation through Temporal Difference (TD) Learning. More importantly, this framework allows us to analyze the limitations of current consistency training/tuning strategies. Built upon Easy Consistency Tuning (ECT), we propose Stable Consistency Tuning (SCT), which incorporates variance-reduced learning using the score identity. SCT leads to significant performance improvements on benchmarks such as CIFAR-10 and ImageNet-64. On ImageNet-64, SCT achieves 1-step FID 2.42 and 2-step FID 1.55, a new SoTA for consistency models.

## 1 Introduction

Diffusion models have significantly advanced the field of visual generation, delivering state-of-the-art performance in images (Dhariwal & Nichol, 2021b; Rombach et al., 2022a; Song & Ermon, 2019; Karras et al., 2022b; 2024b), videos (Shi et al., 2024; Blattmann et al., 2023; Singer et al., 2022; Brooks et al., 2024; Bao et al., 2024), 3D (Gao et al., 2024; Shi et al., 2023), and 4D data (Ling et al., 2024). The core principle of diffusion models is the iterative transformation of pure noise into clean samples. However, this iterative nature comes with a tradeoff: while it enables superior generation quality and training stability compared to traditional methods (Goodfellow et al., 2020; Sauer et al., 2023a), it requires substantial computational resources and longer sampling time (Song et al., 2020a; Ho et al., 2020a). This limitation becomes a substantial bottleneck when generating high-dimensional data, such as high-resolution images and videos, where the increased generation cost slows practical application.

Consistency models (Song et al., 2023a), an emerging generative family, largely address these challenges by enabling high-quality, one-step generation without adversarial training. Recent studies (Song & Dhariwal, 2023; Geng et al., 2024) have shown that one-step and two-step performance of consistency models can rival that of leading diffusion models, which typically require dozens or even hundreds of inference steps, underscoring the tremendous potential of consistency models. The primary training objective of consistency models is to enforce the self-consistency condition (Song et al., 2023a), where predictions for any two points along the same trajectory of the probability flow ODE (PF-ODE) converge to the same solution. To achieve this, consistency models adopt two training methods: consistency distillation (CD) and consistency training/tuning (CT). Consistency distillation leverages a frozen pretrained diffusion model to simulate the PF-ODE, while consistency training/tuning directly learns from real data with no need for extra teacher models.

The motivation of this paper is to propose a novel understanding of consistency models from the perspective of bootstrapping. Specifically, we first frame the numerical solving process of the PF-ODE (i.e., the reverse diffusion process) as a Markov Decision Process (MDP), also indicated in prior works (Black et al., 2023; Fan et al., 2024). The initial state of the MDP is randomly sampled from Gaussian. The intermediate state consists of the denoised sample $\mathbf{x}_t$ and the corresponding

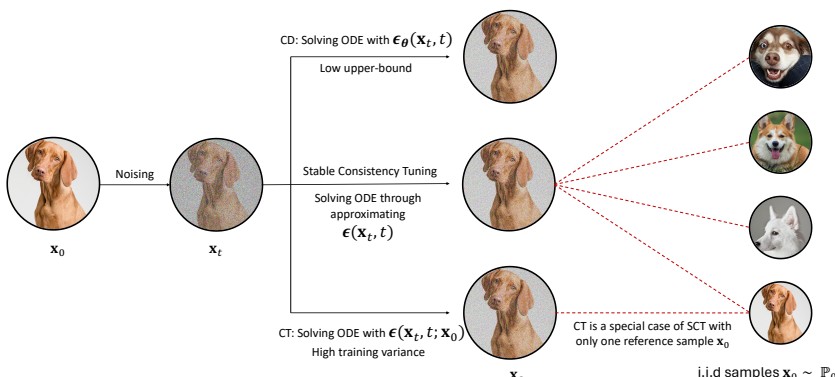

Figure 1: Stable consistency tuning (SCT) with variance reduced training target. SCT provides a unifying perspective to understand different training strategies of consistency models.

conditional information, including the timestep $t$. The policy function of the MDP corresponds to the action of applying the ODE solver to perform single-step denoising, resulting in the transition to the new state. Building on this MDP, we show that the training of consistency models, including consistency distillation, consistency training/tuning, and their variants, can be interpreted as Temporal difference (TD) learning (Sutton & Barto, 2018), with specific reward and value functions aligned with the PF-ODE. From this viewpoint, we can derive, as we will elaborate later, that the key difference between consistency distillation and consistency learning lies in how the ground-truth reward is estimated. The difference leads to distinct behaviors: Consistency distillation has a lower performance upper bound (being limited by the performance of the pretrained diffusion model) but exhibits lower variance and greater training stability. Conversely, consistency training/tuning offers a higher performance upper bound but suffers from a larger variance in reward estimation, which can lead to unstable training. Additionally, for both CD and CT, smaller ODE steps (i.e., $\Delta t = t - r$) can improve the performance ceiling but complicate the optimization.

Building upon the foundation of Easy Consistency Tuning (ECT), we introduce Stable Consistency Tuning (SCT), which incorporates several enhancements for variance reduction and faster convergence: 1) We introduce a variance-reduced training target for consistency training/tuning via the score identity (Vincent, 2011; Xu et al., 2023), which provides a better approximation of the ground truth score. This helps improve training stability and facilitates better performance and convergence. Additionally, we show that variance-reduced estimation can be applied to conditional generation settings for the first time. 2) Our method adopts a smoother progressive training schedule that facilitates training dynamics and reduces discretization error. 3) We extend the scope of ECT to multistep settings, allowing for deterministic multistep sampling. Additionally, we investigate the potential capacity and optimization challenges of multistep consistency models and propose an edge-skipping multistep inference strategy to improve the performance of multistep consistency models. 4) We validate the effectiveness of classifier-free guidance in consistency models, where generation is guided by a sub-optimal version of the consistency model itself.

## 2 PRELIMINARIES ON CONSISTENCY MODELS

In this section, we present the essential background on consistency models to ensure a more self-contained explanation.

**Diffusion Models** define a forward stochastic process with the intermediate distributions $\mathbb{P}_t(\mathbf{x}_t|\mathbf{x}_0)$ conditioned on the initial data $\mathbf{x}_0 \sim \mathbb{P}_0$ (Lipman et al., 2022; Kingma et al., 2021a). The intermediate states follow a general form $\mathbf{x}_t = \alpha_t \mathbf{x}_0 + \sigma_t \epsilon$ with $\mathbf{x}_1 \approx \epsilon \sim \mathcal{N}(0, \mathbf{I})$. The forward process can be described with the following stochastic differential equation (SDE):

$$\mathrm{d}\mathbf{x}_t = f_t \mathbf{x}_0 \mathrm{d}t + g_t \mathrm{d}\boldsymbol{w}_t, \tag{1}$$

where $\boldsymbol{w}_t$ is the standard Wiener process, $f_t = \frac{\mathrm{d}\log\alpha_t}{\mathrm{d}t}$, and $g_t^2 = \frac{\mathrm{d}\sigma_t^2}{\mathrm{d}t} - 2\frac{\mathrm{d}\log\alpha_t}{\mathrm{d}t}\sigma_t^2$. For the above forward SDE, a remarkable property is that there exists a reverse-time ODE trajectory for data

sampling, which is termed as probability flow ODE (PF-ODE) (Song et al., 2023a) That is,

$$d\mathbf{x} = \left[ f_t - \frac{g_t^2}{2} \nabla_\mathbf{x} \log \mathbb{P}_t(\mathbf{x}) \right] dt. \tag{2}$$

It allows for data sampling without introducing additional stochasticity while satisfying the pre-defined marginal distributions $\mathbb{P}_t(\mathbf{x}_t) = \mathbb{E}_{\mathbb{P}(\mathbf{x}_0 | \mathbf{x}_t)}[\mathbb{P}(\mathbf{x}_t \mid \mathbf{x}_0)]$. In diffusion models, a neural network $\phi$ is typically trained to approximate the score function $\mathbf{s}_\phi(\mathbf{x}_t, t) \approx \nabla_\mathbf{x} \log \mathbb{P}_t(\mathbf{x}_t)$, enabling us to apply numerical solver to approximately solving the PF-ODE for sampling. Many works apply epsilon-prediction $\epsilon_\phi(\mathbf{x}_t, t) = -\sigma_t \nabla_\mathbf{x} \log \mathbb{P}_t(\mathbf{x}_t)$ form for training.

**Consistency Models** propose a training approach that teaches the model to directly predict the solution point of the PF-ODE, thus enabling 1-step generation. Specifically, for a given trajectory $\{\mathbf{x}_t\}_{t \in [0,1]}$, the consistency model $\boldsymbol{f}_\theta(\mathbf{x}_t, t)$ is trained to satisfy $\boldsymbol{f}_\theta(\mathbf{x}_t, t) = \mathbf{x}_0, \forall t \in [0, 1]$, where $\mathbf{x}_0$ is the solution point on the same PF-ODE with $\mathbf{x}_t$. The training strategies of consistency models can be categorized into consistency distillation and consistency training. But they share the same training loss design,

$$d(\boldsymbol{f}_\theta(\mathbf{x}_t, t), \boldsymbol{f}_{\theta^-}(\mathbf{x}_r, r)), \tag{3}$$

where $d(\cdot, \cdot)$ is the loss function, $0 \leqslant r < t \leqslant 1$, $\theta^-$ is the EMA weight of $\theta$ or simply set to $\theta$ with gradient disabled for backpropagation. Both $\mathbf{x}_t$ and $\mathbf{x}_r$ should be approximately on the same PF-ODE trajectory.

## 3 UNDERSTANDING CONSISTENCY MODELS

**Consistency model as bootstrapping.** For a general form of diffusion $\mathbf{x}_t = \alpha_t \mathbf{x}_0 + \sigma_t \epsilon$, there exists an exact solution form of PF-ODE as shown in previous work (Song et al., 2021; Lu et al., 2022a),

$$\mathbf{x}_s = \frac{\alpha_s}{\alpha_t} \mathbf{x}_t - \alpha_s \int_{\lambda_t}^{\lambda_s} e^{-\lambda} \epsilon(\mathbf{x}_{t_\lambda}, t_\lambda) d\lambda, \tag{4}$$

where $\lambda_t = \ln(\alpha_t / \sigma_t)$, $t_\lambda$ is the reverse function of $t_\lambda$, and $\epsilon(\mathbf{x}_{t_\lambda, t}) = -\sigma_{t_\lambda} \nabla \log \mathbb{P}_{t_\lambda}(\mathbf{x}_{t_\lambda})$ is the scaled score function. Consistency models aim to learn a $\mathbf{x}_0$ predictor with only the information from $\mathbf{x}_t, \forall t \in [0, 1]$. The left term is already known with $\mathbf{x}_t$, and thereby we can write the consistency model-based $\mathbf{x}_0$ prediction as

$$\hat{\mathbf{x}}_0(\mathbf{x}_t, t; \boldsymbol{\theta}) = \frac{1}{\alpha_t} \mathbf{x}_t - \boldsymbol{h}_\theta(\mathbf{x}_t, t), \tag{5}$$

where $s$ is set to $0$ with $\alpha_s = 1$, $\boldsymbol{\theta}$ is the model weights, and $\boldsymbol{h}_\theta$ is applied to approximate the weighted integral of $\epsilon$ from $t$ to $s = 0$.

The loss of consistency models penalize the $\mathbf{x}_0$ prediction distance between $\mathbf{x}_t$ and $\mathbf{x}_r$ at adjacent timesteps,

$$\hat{\mathbf{x}}_0(\mathbf{x}_t, t; \boldsymbol{\theta}) \xleftarrow{\text{fit}} \hat{\mathbf{x}}_0(\mathbf{x}_r, r; \boldsymbol{\theta}^-), \tag{6}$$

where $0 \leqslant r < t$ and $\boldsymbol{\theta}^-$ is the EMA weighg of $\boldsymbol{\theta}$. Therefore, we have the following learning target

$$\frac{1}{\alpha_t} \mathbf{x}_t - \boldsymbol{h}_\theta(\mathbf{x}_t, t) \xleftarrow{\text{fit}} \frac{1}{\alpha_s} \mathbf{x}_r - \boldsymbol{h}_{\theta^-}(\mathbf{x}_r, r) \tag{7}$$

It is noting that $\mathbf{x}_r = \frac{\alpha_r}{\alpha_t} \mathbf{x}_t - \alpha_r \int_{\lambda_t}^{\lambda_r} e^{-\lambda} \epsilon(\mathbf{x}_{t_\lambda}, t_\lambda) d\lambda$, and hence we replace the $\mathbf{x}_r$ in the above equation and have

$$\boldsymbol{h}_\theta(\mathbf{x}_t, t) \xleftarrow{\text{fit}} \boldsymbol{r} + \boldsymbol{h}_{\theta^-}(\mathbf{x}_r, r), \tag{8}$$

where $\boldsymbol{r} = \int_{\lambda_t}^{\lambda_s} e^{-\lambda} \epsilon(\mathbf{x}_{t_\lambda}, t_\lambda) d\lambda$. The above equation is a Bellman Equation. $\boldsymbol{h}_\theta(\mathbf{x}_t, t)$ is the value estimation at state $\mathbf{x}_t$, $\boldsymbol{h}_{\theta^-}(\mathbf{x}_r, r)$ is the value estimation at state $\mathbf{x}_r$, and $\boldsymbol{r}$ is the step 'reward'.

**Standard formulation.** It is known that the diffusion generation process can be modeled as a Markov Decision Process (MDP), and here we show that the training of consistency models can be viewed

Table 1: The definition of symbols in the value estimation of the PF-ODE equivalent MDP.

| MDP symbols | Definition |
|---|---|
| $s_{t_n}$ | $(t_{N-n}, \mathbf{x}_{t_{N-n}})$ |
| $a_{t_n}$ | $\mathbf{x}_{t_{N-n-1}} := \Phi(\mathbf{x}_{t_{N-n}}, t_{N-n}, t_{N-n-1})$ |
| $P_0(s_0)$ | $(t_N, \mathcal{N}(\mathbf{0}, \mathbf{I}))$ |
| $P(s_{t_{n+1}} \mid s_{t_n}, a_{t_n})$ | $(\delta_{t_{N-n-1}}, \delta_{\mathbf{x}_{t_{N-n-1}}})$ |
| $\pi(a_{t_n} \mid s_{t_n})$ | $\delta_{\mathbf{x}_{t_{N-n-1}}}$ |
| $R(s_{t_n}, a_{t_n})$ | $\int_{\lambda_{t_{N-n}}}^{\lambda_{t_{N-n-1}}} e^{-\lambda} \boldsymbol{\epsilon}(\mathbf{x}_{t_\lambda}, t_\lambda) \mathrm{d}\lambda$ |
| $V_{\boldsymbol{\theta}}(s_{t_n})$ | $\boldsymbol{h}_{\boldsymbol{\theta}}(\mathbf{x}_{t_{N-n}}, t_{N-n})$ |

as a value estimation learning process, which is also known as Temporal Difference Learning (TD-Learning), in the equivalent MDP. We show the standard formulation in Table 1. In Table 1, $s_{t_n}$ and $a_{t_n}$ are the state and action at timestep $t_n$, $P_0$ and $P$ are the initial state distribution and state transition distribution, $\Phi(\mathbf{x}_{t_{N-n}}, t_{N-n}, t_{N-n-1})$ is the ODE solver, $\pi$ is the policy following the PF-ODE, reward $R$ is equivalent to the $r$ defined above and value function $V_{\boldsymbol{\theta}}$ is corresponding to $\boldsymbol{h}_{\boldsymbol{\theta}}$. $\pi$ is the Dirac distribution $\delta$ due to the deterministic nature of PF-ODE.

From this perspective, we can have a unifying understanding of consistency model variants and their behaviors. Fig. 1 provides a straightforward illustration of our insight. One of the most important factors of the consistency model performance is how we estimate $r$ in the equation.

**Understanding consistency distillation.** For consistency distillation, the approximation of $r$ is depent on the pretrained diffusion model $\boldsymbol{\epsilon}_{\boldsymbol{\phi}}$ and the ODE solver applied. For instance, if the first-order DDIM (Song et al., 2020a) is applied, then the approximation is formulated as,

$$\boldsymbol{r} \approx \boldsymbol{\epsilon}_{\boldsymbol{\phi}}(\mathbf{x}_t, t) \int_{\lambda_t}^{\lambda_r} e^{-\lambda} \mathrm{d}\lambda + \mathcal{O}((\lambda_r - \lambda_t)^2). \tag{9}$$

We can observe that the error comes from two aspects: one is the prediction error between the pretrained diffusion model $\boldsymbol{\epsilon}_{\boldsymbol{\phi}}(\mathbf{x}_t, t)$ and the ground truth $\boldsymbol{\epsilon}(\mathbf{x}_t, t)$; the other is the first-order assumption that $\boldsymbol{\epsilon}(\mathbf{x}_{t_\lambda}, t_\lambda) \approx \boldsymbol{\epsilon}(\mathbf{x}_t, t), \forall t_\lambda \in [t, r]$. The first error indicates a better pretrained diffusion model can lead to better performance of consistency distillation. The second error indicates that the distance between $t$ and $s$ should be small eough to eliminate errors caused by low-order approximation. This perspective also connects the n-step TD algorithm with the consistency distllation. The n-step TD is equivalent to apply multistep (n-step) ODE solver to compute the $\mathbf{x}_r$ from $\mathbf{x}_t$.

**Understanding consistency training/tuning.** For consistency training/tuning, the approximation of $r$ is achieved through approximating the groudtruth $\boldsymbol{\epsilon}(\mathbf{x}_t, t)$ with the conditional $\boldsymbol{\epsilon}(\mathbf{x}_t, t; \mathbf{x}_0)$, where $\mathbf{x}_0$ is sampled from the dataset $\mathcal{D}$ and $\mathbf{x}_t = \alpha_t \mathbf{x}_0 + \sigma_t \boldsymbol{\epsilon}$ with $\boldsymbol{\epsilon} \sim \mathcal{N}(\mathbf{0}, \mathbf{I})$. It is konwn that the groudtruth $\boldsymbol{\epsilon}(\mathbf{x}_t, t)$ is equivalent to

$$\begin{aligned}
\boldsymbol{\epsilon}(\mathbf{x}_t, t) &= -\sigma_t \nabla_{\mathbf{x}_t} \log \mathbb{P}_t(\mathbf{x}_t) \\
&= -\sigma_t \mathbb{E}_{\mathbb{P}_t(\mathbf{x}_0 \mid \mathbf{x}_t)} [\nabla_{\mathbf{x}_t} \log \mathbb{P}(\mathbf{x}_t \mid \mathbf{x}_0)] \\
&= -\sigma_t \mathbb{E}_{\mathbb{P}(\mathbf{x}_0 \mid \mathbf{x}_t)} \left[ -\frac{\mathbf{x}_t - \alpha_t \mathbf{x}_0}{\sigma_t^2} \right] = \mathbb{E}_{\mathbb{P}(\mathbf{x}_0 \mid \mathbf{x}_t)} \left[ \frac{\mathbf{x}_t - \alpha_t \mathbf{x}_0}{\sigma_t} \right] = \mathbb{E}_{\mathbb{P}(\mathbf{x}_0 \mid \mathbf{x}_t)} [\boldsymbol{\epsilon}(\mathbf{x}_t, t; \mathbf{x}_0)]
\end{aligned}, \tag{10}$$

where $\boldsymbol{\epsilon}(\mathbf{x}_t, t; \mathbf{x}_0) = \frac{\mathbf{x}_t - \alpha_t \mathbf{x}_0}{\sigma_t}$. In simple terms, the ground truth $\boldsymbol{\epsilon}(\mathbf{x}_t, t)$ is the expectation of all possible conditional epsilon $\boldsymbol{\epsilon}(\mathbf{x}_t, t; \mathbf{x}_0), \forall \mathbf{x}_0 \in \mathcal{D}$. Instead, previous work on consistency training/tuning apply the conditional epsilon to approximate the ground truth epsilon, which can be regarded as a one-shot MCMC approximation. The approximation is formulated as,

$$\boldsymbol{r} \approx \boldsymbol{\epsilon}(\mathbf{x}_t, t; \mathbf{x}_0) \int_{\lambda_t}^{\lambda_r} e^{-\lambda} \mathrm{d}\lambda + \mathcal{O}((\lambda_r - \lambda_t)^2) \tag{11}$$

Similarly, the error comes from two aspects: one is the difference between conditional epsilon $\epsilon(\mathbf{x}_t, t; \mathbf{x}_0)$ and groudtruth epsilon $\epsilon(\mathbf{x}_t, t)$; the other is the first-order approximation error. Even though, it is shown by previous work that the final learning objective will converge to the gourd truth under minor assumptions (e.g, $L$-Lipschitz continuity) (Song et al., 2023a).

$$h_{\boldsymbol{\theta}}(\mathbf{x}_t, t) = \mathbb{E}_{\mathbb{P}(\mathbf{x}_0|\mathbf{x}_t)}\left[\epsilon(\mathbf{x}_t, t; \mathbf{x}_0)\right] \int_{\lambda_t}^{\lambda_r} e^{-\lambda}\mathrm{d}\lambda + h_{\boldsymbol{\theta}}(\mathbf{x}_r, r). \tag{12}$$

However, the variance of one-shot MCMC is large. This causes the consistency training/tuning is not as stable as distillation methods even though it has a better upper bound.

**Summary.** In summary, the main performance bottlenecks in improving consistency training/tuning can be attributed to two factors:

1. **Training Variance**: This refers to the gap between the conditional epsilon $\epsilon(\mathbf{x}_t, t; \mathbf{x}_0)$ and the ground truth epsilon $\epsilon(\mathbf{x}_t, t)$. Although, in theory, the conditional epsilon is expected to match the ground truth epsilon on average, it exhibits higher variance, which introduces instability and deviations during training.

2. **Discretization Error**: In the numerical solving of the ODE for consistency training/tuning, only first-order solvers can be approximated. To push performance to its upper limit, the time intervals between sampled points, $t$ and $r$, must be minimized, i.e., $\mathrm{d}t = \lim(t - r) \to 0$. However, smaller $\mathrm{d}t$ results in a longer information propagation process (with large $N$). If the training process lacks stability, error accumulation through bootstrapping may occur, potentially causing training failure.

## 4 STABLE CONSISTENCY TUNING

Our method builds upon Easy Consistency Tuning (ECT) (Geng et al., 2024), chosen for its efficiency in prototyping. Given our analysis of consistency models from the bootstrapping perspective, we introduce several techniques to enhance performance.

### 4.1 REDUCING THE TRAINING VARIANCE

Previous research has shown that reducing the variance for diffusion training can lead to improved training stability and performance (Xu et al., 2023). However, this technique has only been applied to unconditional generation and diffusion model training. We generalize this technique to both conditional/unconditional generation and consistency training/tuning for variance reduction. Let $\boldsymbol{c}$ represent the conditional inputs (e.g., class labels). We begin with

$$\begin{aligned}
\nabla_{\mathbf{x}_t} \log \mathbb{P}_t(\mathbf{x}_t \mid \boldsymbol{c}) &= \mathbb{E}_{\mathbb{P}(\mathbf{x}_0|\mathbf{x}_t,\boldsymbol{c})}\left[\nabla_{\mathbf{x}_t} \log \mathbb{P}_t(\mathbf{x}_t \mid \mathbf{x}_0, \boldsymbol{c})\right] \\
&= \mathbb{E}_{\mathbb{P}(\mathbf{x}_0|\boldsymbol{c})}\left[\frac{\mathbb{P}(\mathbf{x}_0 \mid \mathbf{x}_t, \boldsymbol{c})}{\mathbb{P}(\mathbf{x}_0 \mid \boldsymbol{c})}\nabla_{\mathbf{x}_t} \log \mathbb{P}_t(\mathbf{x}_t \mid \mathbf{x}_0, \boldsymbol{c})\right] \\
&= \mathbb{E}_{\mathbb{P}(\mathbf{x}_0|\boldsymbol{c})}\left[\frac{\mathbb{P}(\mathbf{x}_t \mid \mathbf{x}_0, \boldsymbol{c})}{\mathbb{P}(\mathbf{x}_t \mid \boldsymbol{c})}\nabla_{\mathbf{x}_t} \log \mathbb{P}_t(\mathbf{x}_t \mid \mathbf{x}_0, \boldsymbol{c})\right] \\
&= \mathbb{E}_{\mathbb{P}(\mathbf{x}_0|\boldsymbol{c})}\left[\frac{\mathbb{P}(\mathbf{x}_t \mid \mathbf{x}_0)}{\mathbb{P}(\mathbf{x}_t \mid \boldsymbol{c})}\nabla_{\mathbf{x}_t} \log \mathbb{P}_t(\mathbf{x}_t \mid \mathbf{x}_0)\right] \\
&\approx \frac{1}{n}\sum_{\substack{i=0,\ldots n-1 \\ \{\mathbf{x}_0^{(i)}\}\sim\mathbb{P}(\mathbf{x}_0|c)}} \frac{\mathbb{P}(\mathbf{x}_t \mid \mathbf{x}_0^{(i)})}{\mathbb{P}(\mathbf{x}_t \mid \boldsymbol{c})}\nabla_{\mathbf{x}_t} \log \mathbb{P}_t(\mathbf{x}_t \mid \mathbf{x}_0^{(i)}) \\
&\approx \frac{1}{n}\sum_{\substack{i=0,\ldots n-1 \\ \{\mathbf{x}_0^{(i)}\}\sim\mathbb{P}(\mathbf{x}_0|c)}} \frac{\mathbb{P}(\mathbf{x}_t \mid \mathbf{x}_0^{(i)})}{\sum_{\mathbf{x}_0^{(j)}\in\{\mathbf{x}_0^{(i)}\}}\mathbb{P}(\mathbf{x}_t \mid \mathbf{x}_0^{(j)}, \boldsymbol{c})}\nabla_{\mathbf{x}_t} \log \mathbb{P}_t(\mathbf{x}_t \mid \mathbf{x}_0^{(i)}) \\
&= \frac{1}{n}\sum_{\substack{i=0,\ldots n-1 \\ \{\mathbf{x}_0^{(i)}\}\sim\mathbb{P}(\mathbf{x}_0|c)}} \frac{\mathbb{P}(\mathbf{x}_t \mid \mathbf{x}_0^{(i)})}{\sum_{\mathbf{x}_0^{(j)}\in\{\mathbf{x}_0^{(i)}\}}\mathbb{P}(\mathbf{x}_t \mid \mathbf{x}_0^{(j)})}\nabla_{\mathbf{x}_t} \log \mathbb{P}_t(\mathbf{x}_t \mid \mathbf{x}_0^{(i)})
\end{aligned} \tag{13}$$

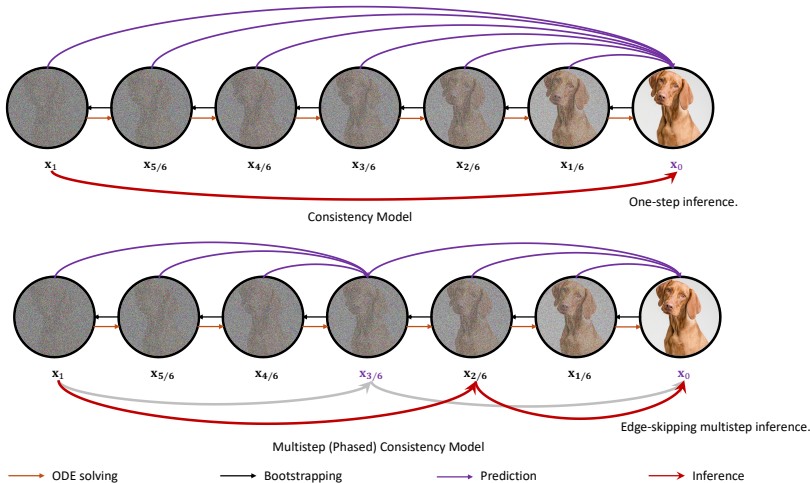

Figure 2: Phasing the ODE path along the time axis for consistency training. We visualize both training and inference techniques in discrete form for easier understanding.

The key difference between the variance-reduced score estimation of conditional generation and unconditional generation is whether the samples utilized for computing the variance-reduced target are sampled from the conditional distribution $\mathbb{P}(\mathbf{x}_0 \mid \boldsymbol{c})$ or not. In the class-conditional generation, this means we compute stable targets only within each class cluster. For text-to-image generation, we might estimate probabilities using CLIP (Radford et al., 2021) text-image similarity, though we leave this for future study. Therefore, the conditional epsilon estimation adopted in previous consistency training/tuning can be replaced by our variance-reduced estimation:

$$
\begin{aligned}
\boldsymbol{\epsilon}(\mathbf{x}_t, t) &= -\sigma_t \nabla_{\mathbf{x}_t} \log \mathbb{P}_t(\mathbf{x}_t) \\
&\approx \frac{1}{n} \sum_{\substack{i=0,\ldots n-1 \\ \{\mathbf{x}_0^{(i)}\} \sim \mathbb{P}(\mathbf{x}_0|c)}} \frac{\mathbb{P}(\mathbf{x}_t \mid \mathbf{x}_0^{(i)})}{\sum_{\mathbf{x}_0^{(j)} \in \{\mathbf{x}_0^{(i)}\}} \mathbb{P}(\mathbf{x}_t \mid \mathbf{x}_0^{(j)})} (-\sigma_t \nabla_{\mathbf{x}_t} \log \mathbb{P}_t(\mathbf{x}_t \mid \mathbf{x}_0^{(i)})) \\
&= \frac{1}{n} \sum_{i=0}^{n-1} W_i \boldsymbol{\epsilon}(\mathbf{x}_t, t; \mathbf{x}_0^{(i)})) ,
\end{aligned}
\tag{14}
$$

where $W_i = \frac{\mathbb{P}(\mathbf{x}_t|\mathbf{x}_0^{(i)})}{\sum_{\mathbf{x}_0^{(j)} \in \{\mathbf{x}_0^{(i)}\}} \mathbb{P}(\mathbf{x}_t|\mathbf{x}_0^{(j)})}$ is the weight of conditional $\boldsymbol{\epsilon}(\mathbf{x}_t, t; \mathbf{x}_0^{(i)})$.

## 4.2 REDUCING THE DISCRETIZATION ERROR

As discussed earlier, to achieve higher performance, we need to minimize $\Delta t = (t - r)$. On one hand, when $\Delta t$ is relatively large, the model suffers from increased discretization errors. On the other hand, when $\Delta t$ is too small, it may lead to error accumulation or even training failure. Previous works (Song et al., 2023a; Song & Dhariwal, 2023; Geng et al., 2024) employ a progressive training strategy, which has consistently been shown to be effective. The model is initially trained with a relatively large $\Delta t$, and as training progresses, $\Delta t$ is gradually reduced. Although a larger $\Delta t$ introduces higher discretization errors, it allows for faster optimization, enabling the model to quickly learn a coarse solution. Gradually decreasing $\Delta t$ allows the model to learn more fine-grained results, ultimately improving performance. In the ECT, the training schedule is determined by

$$
t \sim \text{LogNormal}(P_{\text{mean}}, P_{\text{std}}), \quad r := \text{ReLU}\left(1 - \frac{1}{q^{\lceil \text{iter}/d \rceil}} n(t)\right) t
\tag{15}
$$

, where $q$ is to determine the shrinking speed, $d$ is to determine the shrinking frequency, ReLU is equivalent to $\max(\cdot, 0)$, and $n(t)$ is a pre-defined monotonic function. We note that it is beneficial to apply a smoother shrinking process. That is, we reduce both $q$ and $d$ to obtain a smoother shrinking process than the original ECT settings. This provides our method with a faster and smoother training

process. In addition to the training schedule, training weight is important to balance the training across different timesteps. We apply the weighting $1/(t - r)$ following previous work (Song & Dhariwal, 2023; Geng et al., 2024). Suppose $r = \alpha t$, the weighting can be decomposed into $\frac{1}{t} \times \frac{1}{(1-\alpha)}$. The weighting scheme has two key effects: First, $1/t$ assigns higher weights to smaller timesteps, where uncertainty is lower. Predictions at smaller timesteps serve as teacher models for larger timesteps, making stable training at these smaller steps crucial. Second, $1/(1 - \alpha)$ ensures that as $\Delta t$ decreases, the weight dynamically increases, preventing gradient vanishing during training. We apply a smooth term $\delta > 0$ in the weighting function $1/(t - r + \delta) \leqslant \frac{1}{\delta}$ to avoid potential numerical issues and instability when the $\Delta t$ becomes too tiny.

### 4.3 PHASING THE ODE FOR CONSISTENCY TUNING

Previous works (Heek et al., 2024; Wang et al., 2024a) propose dividing the ODE path along the time axis into multiple segments during training, enabling consistency models to support deterministic multi-step sampling with improved performance. We test our method in this scenario and find that, while this training approach increases the minimum required sampling steps, it improves the fidelity and stability of the generated results. We apply the Euler solver to achieve multistep re-parameterization, formulated as:

$$\mathbf{x}_s = D_{\boldsymbol{\theta}}(\mathbf{x}_t) + \frac{s}{t}(\mathbf{x}_t - D_{\boldsymbol{\theta}}(\mathbf{x}_t)), \tag{16}$$

where $D_{\boldsymbol{\theta}}$ denotes the original consistency model, predicting the ODE solution point $\mathbf{x}_0$, and $s$ is the edge timestep. We propose a new training schedule to adapt to the multistep training setting.

$$t \sim \text{LogNormal}(P_{\text{mean}}, P_{\text{std}}), \quad r := \text{ReLU}\left(1 - \frac{1}{q^{\lceil \text{iter}/d \rceil}} n(t)\right)(t - s) + s \tag{17}$$

### 4.4 EXPLORING BETTER INFERENCE FOR CONSISTENCY MODEL

**Guiding consistency models with a bad version of itself.** Previous work (Karras et al., 2024a) demonstrates that even unconditional diffusion models can benefit from classifier-free guidance (Ho & Salimans, 2022). It suggests that the unconditional outputs in classifier-free guidance can be replaced with outputs from a sub-optimal version of the same diffusion model, thus extending the applicability of classifier-free guidance.

$$\nabla_{\mathbf{x}_t} \log \mathbb{P}_{\boldsymbol{\theta}}(\mathbf{x}_t | \boldsymbol{c}; t) + \nabla_{\mathbf{x}_t} \log \left[\frac{\mathbb{P}_{\boldsymbol{\theta}}(\mathbf{x}_t | \boldsymbol{c}; t)}{\mathbb{P}_{\boldsymbol{\theta}^\star}(\mathbf{x}_t | \boldsymbol{c}; t)}\right]^{\omega}, \tag{18}$$

where $\omega$ is the guidance strength, $\boldsymbol{\theta}^\star$ is a sub-optimal version of $\boldsymbol{\theta}$, and $\boldsymbol{c}$ represents the optional label conditions. Our empirical investigations confirm that this strategy can be applied to consistency models, resulting in enhanced sample quality.

**Edge-skipping inference for multistep consistency model.** While segmenting the ODE path to train a multistep consistency model can enhance generation quality, it may encounter optimization challenges, especially around the edge timesteps $\{s_i\}_{i=1}^n$ with $s_1 = 1 > \cdots > s_i > \cdots > s_n = 0$. For timesteps $s_{i-1} \geqslant t > s_i$, the consistency model learns to predict $\mathbf{x}_{s_i}$ from $\mathbf{x}_t$. However, for $s_i \geqslant t' > s_{i+1}$, the model learns to predict $\mathbf{x}_{s_{i+1}}$ from $\mathbf{x}_{t'}$. When $t$ and $t'$ are very close to $s_i$, denoted as $t = s_i^+$ and $t' = s_i^-$, it is apparent that $\mathbf{x}_{s_i^+}$ and $\mathbf{x}_{s_i^-}$ can be very similar. However, the model is expected to predict two distinct results ($\mathbf{x}_{s_i}$ and $\mathbf{x}_{s_{i+1}}$) from very similar inputs ($\mathbf{x}_{s_i^+}$ and $\mathbf{x}_{s_i^-}$).

Neural networks typically follow $L$-Lipschitz continuity, where small input changes result in small output changes. This property conflicts with the requirement to produce distinct outputs from similar inputs near edge timesteps, potentially leading to insufficient training, particularly near $s_i^-$. To address this, we propose skipping the edge timesteps during multistep sampling. Specifically, even though we aim for the model to perform sampling through the timesteps

$$s_1 := 1 \rightarrow s_2 \rightarrow s_3 \rightarrow \cdots \rightarrow s_n := 0, \tag{19}$$

we instead achieve multistep sampling via

$$s_1 := 1 \rightarrow \eta s_2 \rightarrow \eta s_3 \rightarrow \cdots \rightarrow \eta s_n := 0, \tag{20}$$

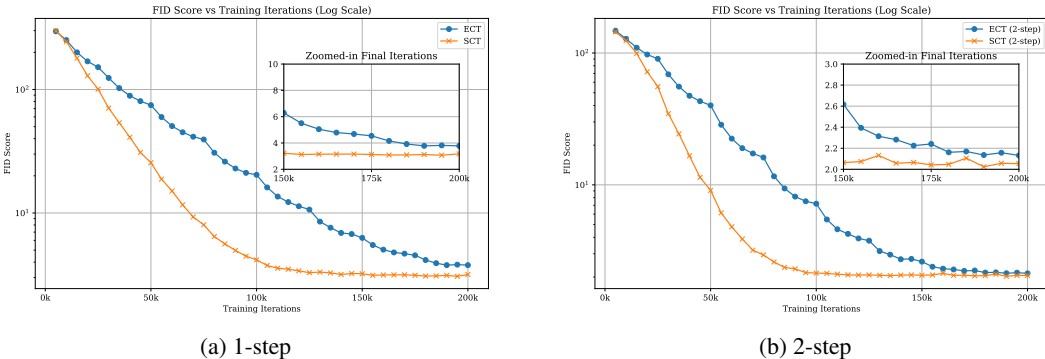

(a) 1-step                                    (b) 2-step

Figure 3: FID vs Training iterations. SCT has faster convergence speed and better performance upper bound than ECT.

where $\eta > 0$ is a scaling factor. When $\eta$ is set to 1, the process reverts to normal multistep sampling. This method works because the predictions of $\mathbf{x}_{s_i}$ and $\mathbf{x}_{\eta s_i}$ are close when $\eta$ is near 1, allowing for a tolerable degree of approximation error. Fig. 2 illustrates this concept with a discrete example. The model is designed to sample via the sequence $\mathbf{x}_1 \to \mathbf{x}_{3/6} \to \mathbf{x}_0$; however, it instead samples through the sequence $\mathbf{x}_1 \to \mathbf{x}_{2/6} \to \mathbf{x}_0$.

## 5 EXPERIMENTS

### 5.1 EXPERIMENT SETUPS

**Evaluation Benchmarks.** Following the evaluation protocols of iCT (Song & Dhariwal, 2023) and ECT (Geng et al., 2024), we validate the effectiveness of SCT on CIFAR-10 (unconditional and conditional) (Krizhevsky et al., 2009) and ImageNet-64 (conditional) (Deng et al., 2009). Performance is measured using Frechet Inception Distance (FID, lower is better) (Heusel et al., 2017) consistent with recent studies (Geng et al., 2024; Karras et al., 2024b).

**Compared baselines.** We compare our method against accelerated samplers (Lu et al., 2022a; Zhao et al., 2024), state-of-the-art diffusion-based methods (Ho et al., 2020a; Song & Ermon, 2019; 2020; Karras et al., 2022b), distillation methods (Zhou et al., 2024; Salimans & Ho, 2022a), alongside consistency training and tuning approaches. Among these models, consistency training and tuning methods serve as key baselines, including CT (LIPIPS) (Song et al., 2023a), iCT (Song & Dhariwal, 2023), ECT (Geng et al., 2024), and MCM (CT) (Heek et al., 2024). CT introduces the first consistency training algorithm, utilizing LIPIPS loss to improve FID performance. iCT presents an improved training strategy over CT, making the performance of consistency training comparable to state-of-the-art diffusion models for the first time. MCM (CT) proposes segmenting the ODE path for consistency training, while ECT introduces the concept of consistency tuning along with a continuous-time training strategy, achieving notable results with significantly reduced training costs.

**Model Architectures and Training Configurations** From a model perspective, iCT is based on the ADM (Dhariwal & Nichol, 2021b), ECT is built on EDM2 (Karras et al., 2024b), and MCM follows the UViTs of Simple Diffusion (Hoogeboom et al., 2023). The model size of ECT is similar to that of iCT, while MCM does not explicitly specify the model size. The iCT model is randomly initialized, whereas both ECT and MCM use pretrained diffusion models for initialization. In terms of training costs, iCT uses a batch size of 4096 across 800,000 iterations, MCM employs a batch size of 2048 for 200,000 iterations, and ECT utilizes a batch size of 128 for 100,000 iterations. SCT follows ECT's model architecture and training configuration.

### 5.2 RESULTS AND ANALYSIS

**Training efficiency and efficacy.** In Fig. 3b, we plot 1-step FID and 2-step FID for SCT and ECT along the number of training epochs, under the same training configuration. From the figure, we observe that SCT significantly improves convergence speed compared to ECT, demonstrating the

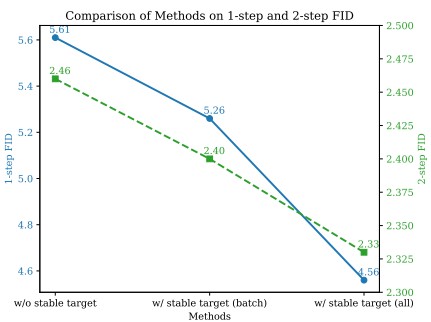 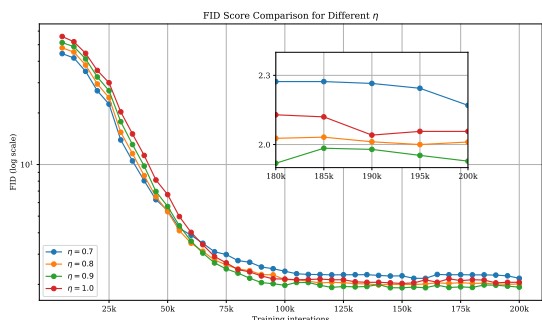

Figure 4: The effectiveness of variance reduced training target.

Figure 5: The effectiveness of edge-skipping multi-step sampling.

efficiency and efficacy of SCT training. Additionally, the performance comparisons in Tables 2 and 3 also show that SCT outperforms ECT across different settings.

**Quantitative evaluation.** We present results in Table 2 and Table 3. Our approach consistently outperforms ECT across various scenarios, achieving results comparable to advanced distillation strategies and diffusion/score-based models.

**The effectiveness of training variance reduction.** It is worth noting that SCT and ECT employ different progressive training schedules. To exclude this effect, we adopt ECT's fixed training schedule, in which the 2-step FID surpasses Consistency Distillation within a single A100 GPU hour. We use $\Delta t = t/256$ as a fixed partition, with a batch size of 128, over 16k iterations on CIFAR-10, while keeping all other settings unchanged. For SCT models on CIFAR-10, we calculate the variance-reduced target only within the training batch, which is also the default setting of all our experiments on CIFAR-10. To further showcase the effectiveness of the variance-reduced target, we use all 50,000 train-

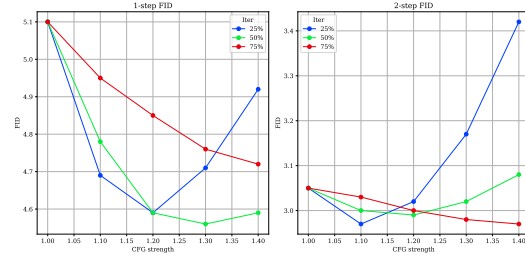

Figure 6: The effectiveness of classifier-free guidance on consistency models.

ing samples as a reference to compute the target. Although more reference samples are used, they do not directly influence the model's computations; they are solely utilized for calculating the training target. Fig. 4 presents a comparison of these three methods, showing that our approach achieves notable improvements in both 1-step and 2-step FID. Notably, when using the entire sample set as the reference batch, the improvement becomes more pronounced, with the 1-step FID dropping from 5.61 to 4.56.

**The Effectiveness of CFG.** Inspired by prior work Karras et al. (2024a), we adopt the outputs of the sub-optimal version of the model as the negative part in classifier-free guidance (CFG). We set the CFG strength as 1.2 and the sub-optimal version as the ema weight with half training iterations by default. We investigate the influence of the two factors on SCT-S models on ImageNet. As illustrated in Fig. 6, an appropriate CFG setting can significantly enhance the quality of generation.

**Edge-skipping Multistep Sampling.** To demonstrate the effectiveness of our method, we record the 4-step FID curve at various training stages, utilizing different $\eta$ values for edge-skipping multistep inference. We find that a smaller $\eta$ at the beginning of training yields superior performance. As training progresses, the model's estimates of multi-stage results become increasingly accurate, and larger $\eta$ values gradually enhance performance. However, as previously analyzed, the multistep model struggles to achieve perfect multistep training, leading to better overall performance for $\eta = 0.9$ compared to $\eta = 1.0$ (the default method).

**Sample Quality of SCT.** We showcase the generation results of SCT models in Fig. 7, Fig. 8, Fig. 9, Fig. 10, Fig. 11, Fig. 12, Fig. 13, Fig. 14, and Fig. 15. The majority of generated samples show favorable low-frequency compositions and high-frequency details.

Table 2: Comparing the quality of samples on CIFAR-10.

| METHOD | NFE ($\downarrow$) | FID ($\downarrow$) |
|---|---|---|
| **Fast samplers & distillation for diffusion models** | | |
| DDIM (Song et al., 2020b) | 10 | 13.36 |
| DPM-solver-fast (Lu et al., 2022b) | 10 | 4.70 |
| 3-DEIS (Zhang & Chen, 2022) | 10 | 4.17 |
| UniPC (Zhao et al., 2024) | 10 | 3.87 |
| Knowledge Distillation (Luhman & Luhman, 2021) | 1 | 9.36 |
| DFNO (LPIPS) (Zheng et al., 2022) | 1 | 3.78 |
| 2-Rectified Flow (+distill) (Liu et al., 2022) | 1 | 4.85 |
| TRACT (Berthelot et al., 2023) | 1 | 3.78 |
| | 2 | 3.32 |
| Diff-Instruct (Luo et al., 2023) | 1 | 4.53 |
| PD (Salimans & Ho, 2022b) | 1 | 8.34 |
| | 2 | 5.58 |
| CTM (Kim et al., 2023) | 1 | 5.19 |
| | 18 | 3.00 |
| CTM (+GAN +CRJ) | 1 | 1.98 |
| | 2 | 1.87 |
| SiD ($\alpha = 1.0$) (Zhou et al., 2024) | 1 | 2.03 |
| SiD ($\alpha = 1.2$) (Zhou et al., 2024) | 1 | 1.98 |
| CD (LPIPS) (Song et al., 2023b) | 1 | 3.55 |
| | 2 | 2.93 |
| **Direct Generation** | | |
| Score SDE (Song et al., 2021) | 2000 | 2.38 |
| Score SDE (deep) (Song et al., 2021) | 2000 | 2.20 |
| DDPM (Ho et al., 2020b) | 1000 | 3.17 |
| LSGM (Vahdat et al., 2021) | 147 | 2.10 |
| PFGM (Xu et al., 2022) | 110 | 2.35 |
| EDM (Karras et al., 2022a) | 35 | 2.04 |
| EDM-G++ (Kim et al., 2022) | 35 | 1.77 |
| NVAE (Vahdat & Kautz, 2020) | 1 | 23.5 |
| Glow (Kingma & Dhariwal, 2018) | 1 | 48.9 |
| Residual Flow (Chen et al., 2019) | 1 | |
| BigGAN (Brock et al., 2019) | 1 | 14.7 |
| StyleGAN2 (Karras et al., 2020b) | 1 | 8.32 |
| StyleGAN2-ADA (Karras et al., 2020a) | 1 | 2.92 |
| **Consistency Training/Tuning** | | |
| CT (LPIPS) (Song et al., 2023b) | 1 | 8.70 |
| | 2 | 5.83 |
| iCT (Song & Dhariwal, 2023) | 1 | 2.83 |
| | 2 | 2.46 |
| iCT-deep (Song & Dhariwal, 2023) | 1 | 2.51 |
| | 2 | 2.24 |
| ECT (Geng et al., 2024) | 1 | 3.78 |
| | 2 | 2.13 |
| SCT | 1 | 3.11 (2.98) |
| | 2 | 2.05 (2.05) |
| SCT$^\star$ | 1 | 2.92 (2.78) |
| | 2 | 2.02 (1.94) |
| SCT (Phased) | 4 | 1.95 |
| | 8 | 1.86 |
| Cond-SCT | 1 | 3.03 (2.94) |
| | 2 | 1.88 (1.86) |
| Cond-SCT$^\star$ | 1 | 2.88 (2.82) |
| | 2 | 1.87 (1.84) |

Table 3: Comparing the quality of class-conditional samples on ImageNet-64.

| METHOD | NFE ($\downarrow$) | FID ($\downarrow$) |
|---|---|---|
| **Fast samplers & distillation for diffusion models** | | |
| DDIM (Song et al., 2020b) | 50 | 13.7 |
| | 10 | 18.3 |
| DPM solver (Lu et al., 2022b) | 10 | 7.93 |
| | 20 | 3.42 |
| DEIS (Zhang & Chen, 2022) | 10 | 6.65 |
| | 20 | 3.10 |
| DFNO (LPIPS) (Zheng et al., 2022) | 1 | 7.83 |
| TRACT (Berthelot et al., 2023) | 1 | 7.43 |
| | 2 | 4.97 |
| BOOT (Gu et al., 2023) | 1 | 16.3 |
| Diff-Instruct (Luo et al., 2023) | 1 | 5.57 |
| PD (Salimans & Ho, 2022b) | 1 | 15.39 |
| | 2 | 8.95 |
| | 4 | 6.77 |
| CTM (+GAN + CRJ) (Kim et al., 2023) | 1 | 1.92 |
| SiD ($\alpha = 1.0$) (Zhou et al., 2024) | 1 | 2.03 |
| PD (LPIPS) (Song et al., 2023b) | 1 | 7.88 |
| | 2 | 5.74 |
| | 4 | 4.92 |
| CD (LPIPS) (Song et al., 2023b) | 1 | 6.20 |
| | 2 | 4.70 |
| | 3 | 4.32 |
| **Direct Generation** | | |
| RIN (Jabri et al., 2022) | 1000 | 1.23 |
| DDPM (Ho et al., 2020b) | 250 | 11.0 |
| iDDPM (Nichol & Dhariwal, 2021) | 250 | 2.92 |
| ADM (Dhariwal & Nichol, 2021a) | 250 | 2.07 |
| EDM (Karras et al., 2022a) | 511 | 1.36 |
| EDM* (Heun) (Karras et al., 2022a) | 79 | 2.44 |
| BigGAN-deep (Brock et al., 2019) | 1 | 4.06 |
| **Consistency Training/Tuning** | | |
| CT (LPIPS) (Song et al., 2023b) | 1 | 13.0 |
| | 2 | 11.1 |
| iCT (Song & Dhariwal, 2023) | 1 | 4.02 |
| | 2 | 3.20 |
| iCT-deep (Song & Dhariwal, 2023) | 1 | 3.25 |
| | 2 | 2.77 |
| MCM (CT) (Heek et al., 2024) | 1 | 7.2 |
| | 2 | 2.7 |
| | 4 | 1.8 |
| ECT-S (Geng et al., 2024) | 1 | 5.51 |
| | 2 | 3.18 |
| ECT-M (Geng et al., 2024) | 1 | 3.67 |
| | 2 | 2.35 |
| ECT-XL (Geng et al., 2024) | 1 | 3.35 |
| | 2 | 1.96 |
| SCT-S | 1 | 5.10 (4.59) |
| | 2 | 3.05 (2.98) |
| | 4 | 2.51 (2.43) |
| SCT-M | 1 | 3.30 (3.06) |
| | 2 | 2.13 (2.09) |
| | 4 | 1.83 (1.78) |
| SCT-M$^\star$ | 1 | 2.42 (2.23) |
| | 2 | 1.55 (1.47) |

Results for existing methods are taken from a previous papers. Results of SCT on CIFAR-10 without $\star$ are trained with batch size 128 for 200k iterations. Results of SCT on CIFAR-10 with $\star$ are trained with batch size 512 for 300k iterations. Results of SCT on ImageNet-64 without $\star$ are trained with batch size 128 for 100k iterations. Results of SCT on ImageNet-64 with $\star$ are trained with batch size 1024 for 100k iterations. The metrics inside the parentheses were obtained using CFG. CTM applies classifier rejection sampling (CRJ) for better FID, which needs to generate more samples than other methods.

# 6 CONCLUSION

In this work, we propose Stable Consistency Tuning (SCT), a novel approach that unifies and improves consistency models. By addressing the challenges in training variance and discretization errors, SCT achieves faster convergence and offers insights for further improvements. Our experiments demonstrate state-of-the-art 1-step and few-step generative performance on both CIFAR-10 and ImageNet-64$\times$64, offering a new perspective for future studies on consistency models.

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

# APPENDIX

## I  RELATED WORKS

**Diffusion Models.** Diffusion models (Ho et al., 2020a; Song et al., 2021; Karras et al., 2022b) have emerged as leading foundational models in image synthesis. Recent studies have developed their theoretical foundations (Lipman et al., 2022; Chen & Lipman, 2023; Song et al., 2021; Kingma et al., 2021b) and sought to expand and improve the sampling and design space of these models (Song et al., 2020a; Karras et al., 2022b; Kingma et al., 2021b). Other research has explored architectural innovations for diffusion models (Dhariwal & Nichol, 2021b; Peebles & Xie, 2023), while some have focused on scaling these models for text-conditioned image synthesis and various real-world applications (Shi et al., 2024; Rombach et al., 2022b; Podell et al., 2023). Efforts to accelerate the sampling process include approaches at the scheduler level (Karras et al., 2022b; Lu et al., 2022a; Song et al., 2020a) and the training level (Meng et al., 2023; Song et al., 2023a), with the former often aiming to improve the approximation of the probability flow ODE (Lu et al., 2022a; Song et al., 2020a). The latter primarily involves distillation techniques (Meng et al., 2023; Salimans & Ho, 2022a) or initializing diffusion model weights for GAN training (Sauer et al., 2023b; Lin et al., 2024).

**Consistency Models.** Consistency models are an emerging class of generative models (Song et al., 2023a; Song & Dhariwal, 2023) for fast high-quality generation. It can be trained through either consistency distillation or consistency training. Advanced methods have demonstrated that consistency training can surpass diffusion model training in performance (Song & Dhariwal, 2023; Geng et al., 2024). Several studies propose different strategies for segmenting the ODE (Kim et al., 2023; Heek et al., 2024; Wang et al., 2024a), while others explore combining consistency training with GANs to enhance training efficiency (Kong et al., 2024). Additionally, the consistency model framework has been applied to video generation (Wang et al., 2024b; Mao et al., 2024), language modeling (Kou et al., 2024) and policy learning (Prasad et al., 2024).

## II  LIMITATIONS

The work is limited to traditional benchmarks with CIFAR-10 and ImageNet-64 to validate the effectiveness of unconditional generation and class-conditional generation. However, previous works, including iCT (Song & Dhariwal, 2023) and ECT (Geng et al., 2024), only validate their effectiveness on these two benchmarks. We hope future research explores consistency training/tuning at larger scales such as text-to-image generation.

## III  QUALITATIVE RESULTS

864
865
866
867
868
869
870
871
872
873
874
875
876
877
878
879
880
881
882
883
884
885
886
887
888
889
890
891
892
893
894
895
896
897
898
899
900
901
902
903
904
905
906
907
908
909
910
911
912
913
914
915
916
917

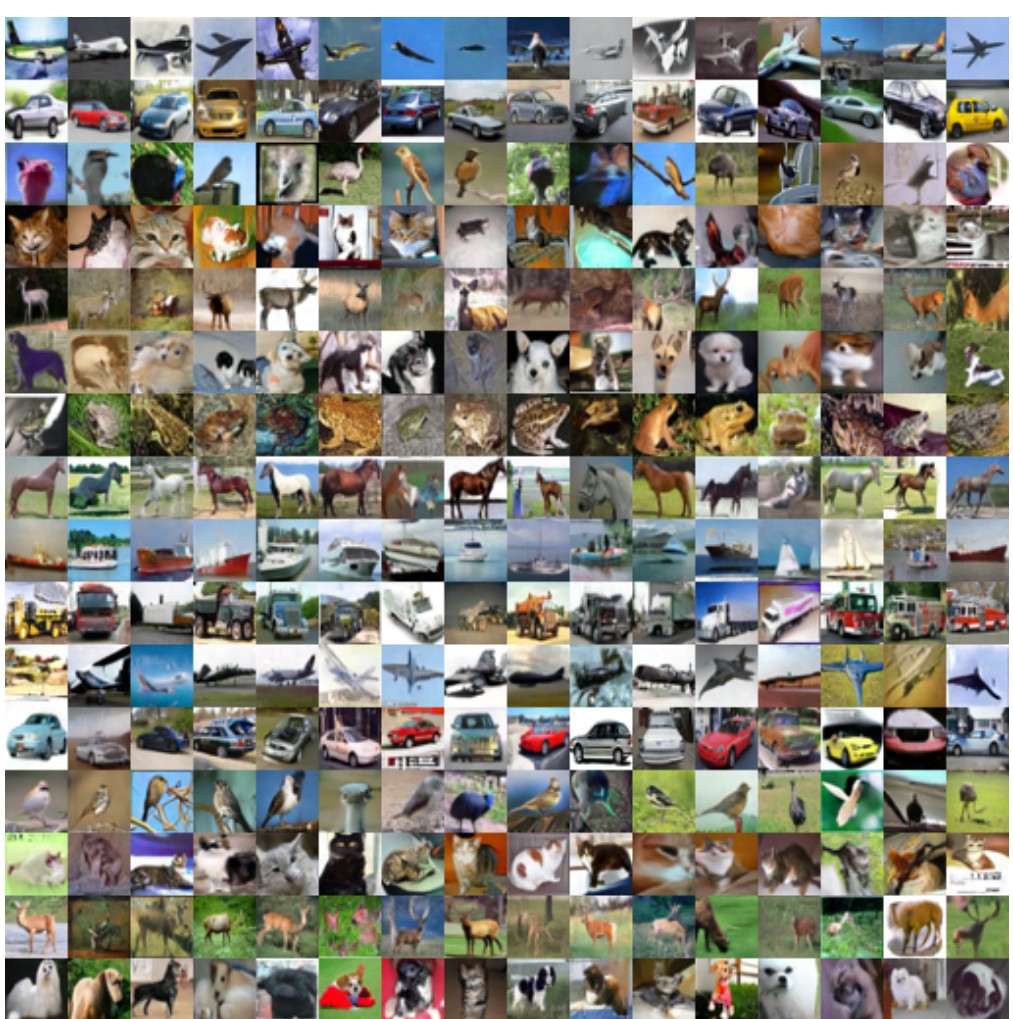

Figure 7: 1-step samples from class-conditional SCT trained on CIFAR-10. Each row corresponds to a different class.

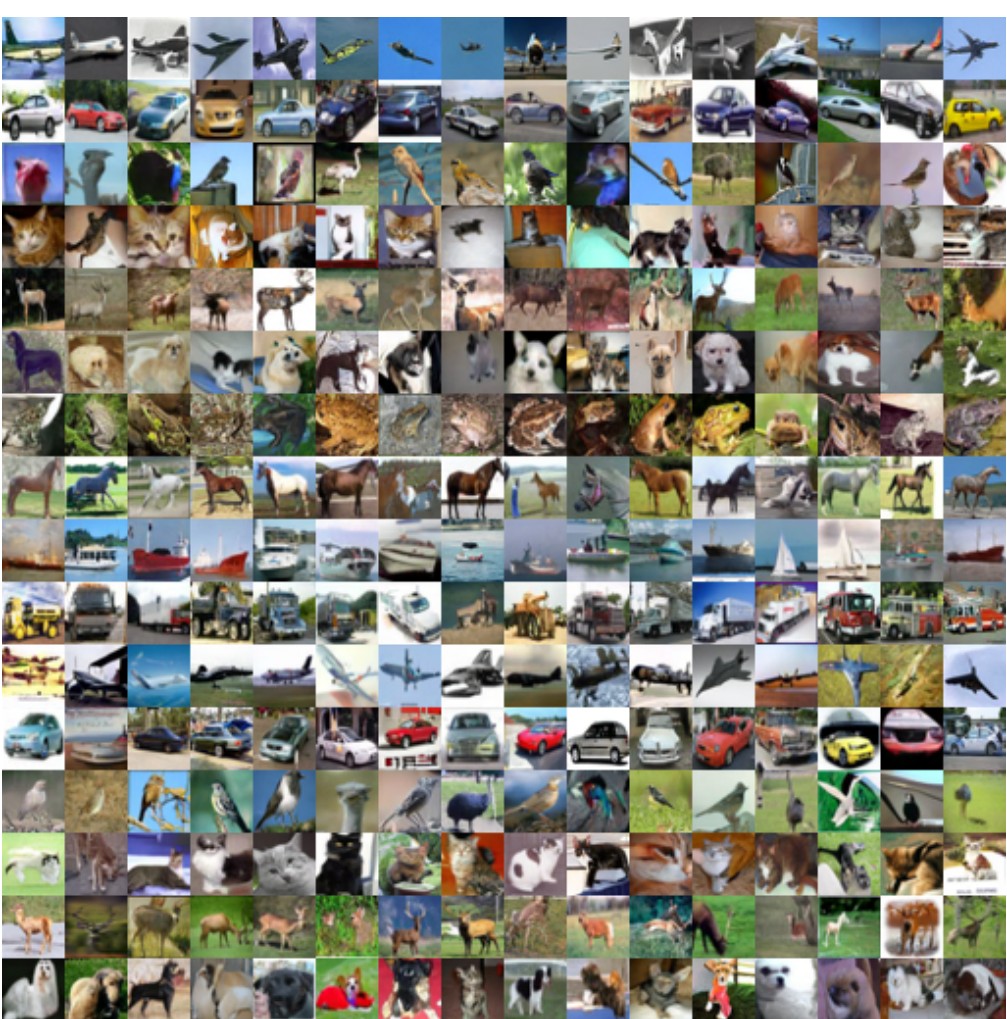

Figure 8: 2-step samples from class-conditional SCT trained on CIFAR-10. Each row corresponds to a different class.

972
973
974
975
976
977
978
979
980
981
982
983
984
985
986
987
988
989
990
991
992
993
994
995
996
997
998
999
1000
1001
1002
1003
1004
1005
1006
1007
1008
1009
1010
1011
1012
1013
1014
1015
1016
1017
1018
1019
1020
1021
1022
1023
1024
1025

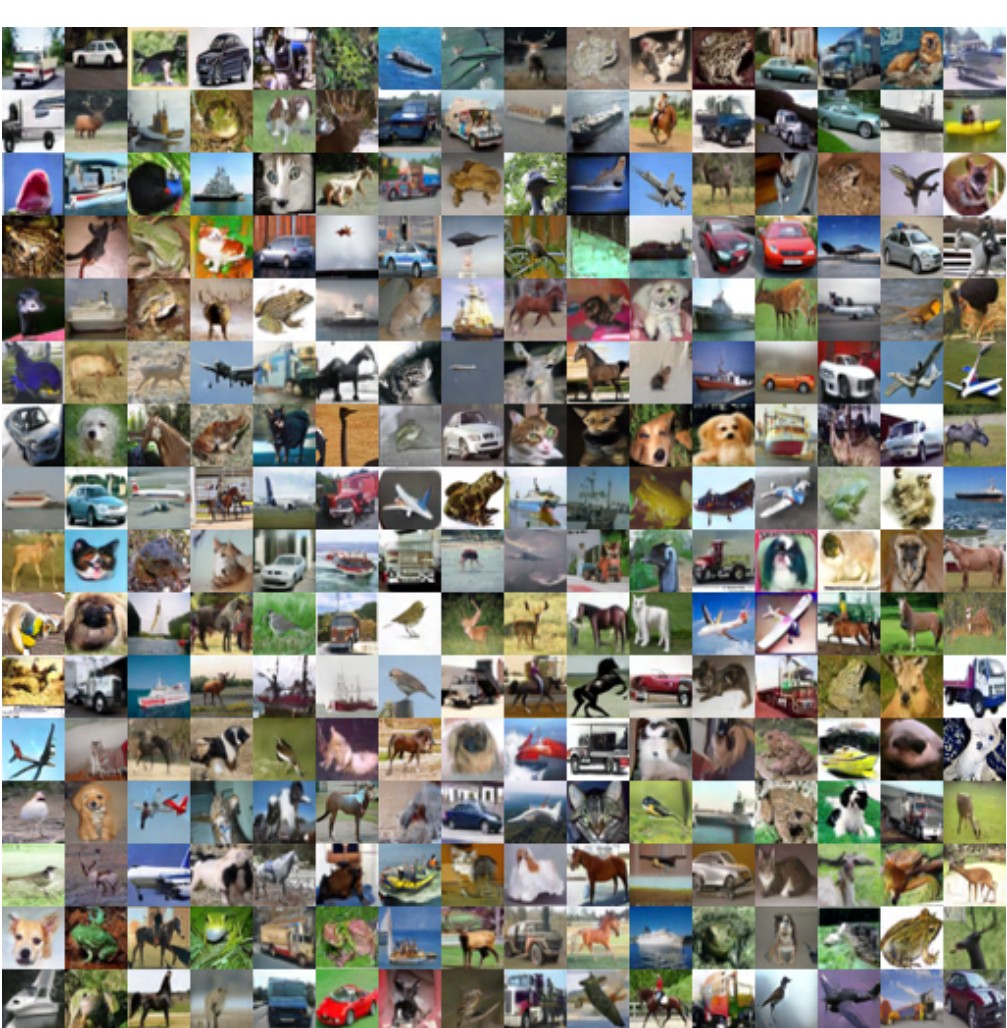

Figure 9: 1-step samples from unconditional SCT trained on CIFAR-10.

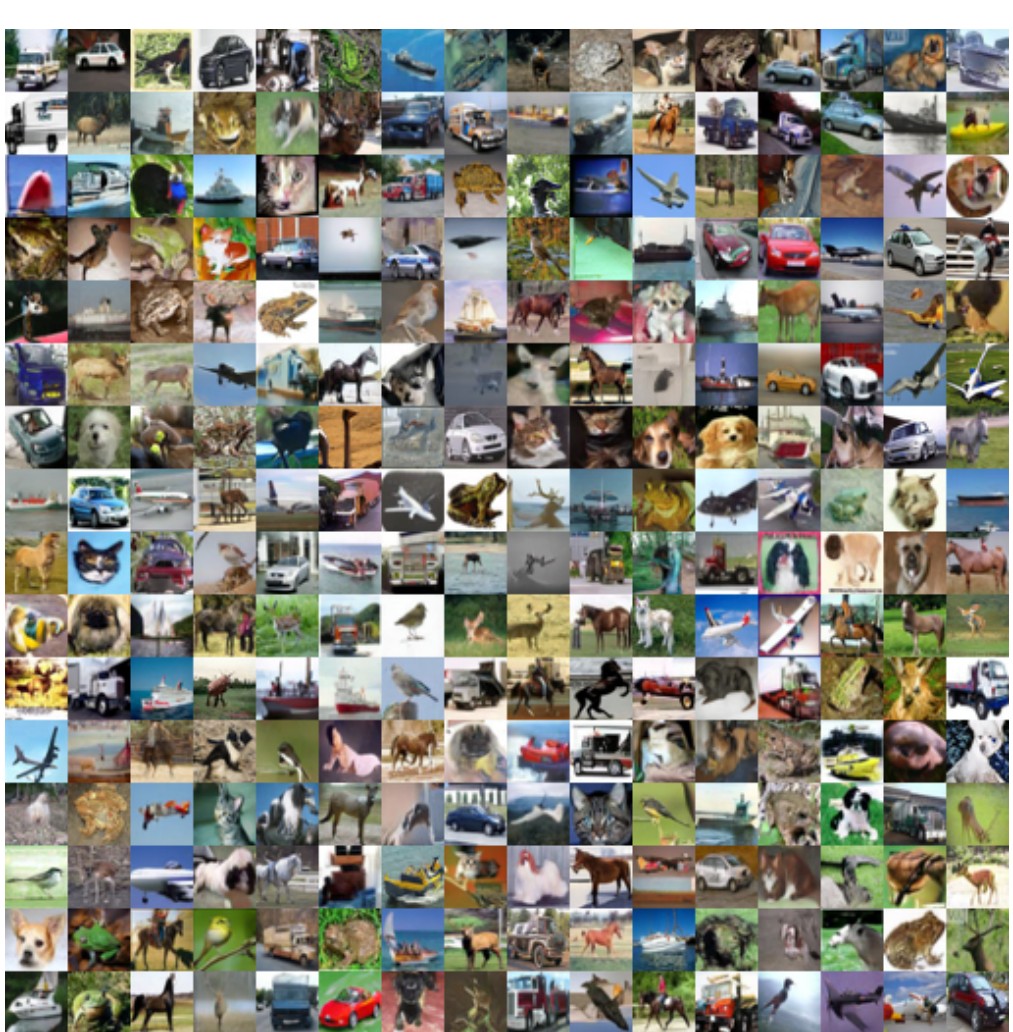

Figure 10: 2-step samples from unconditional SCT trained on CIFAR-10.

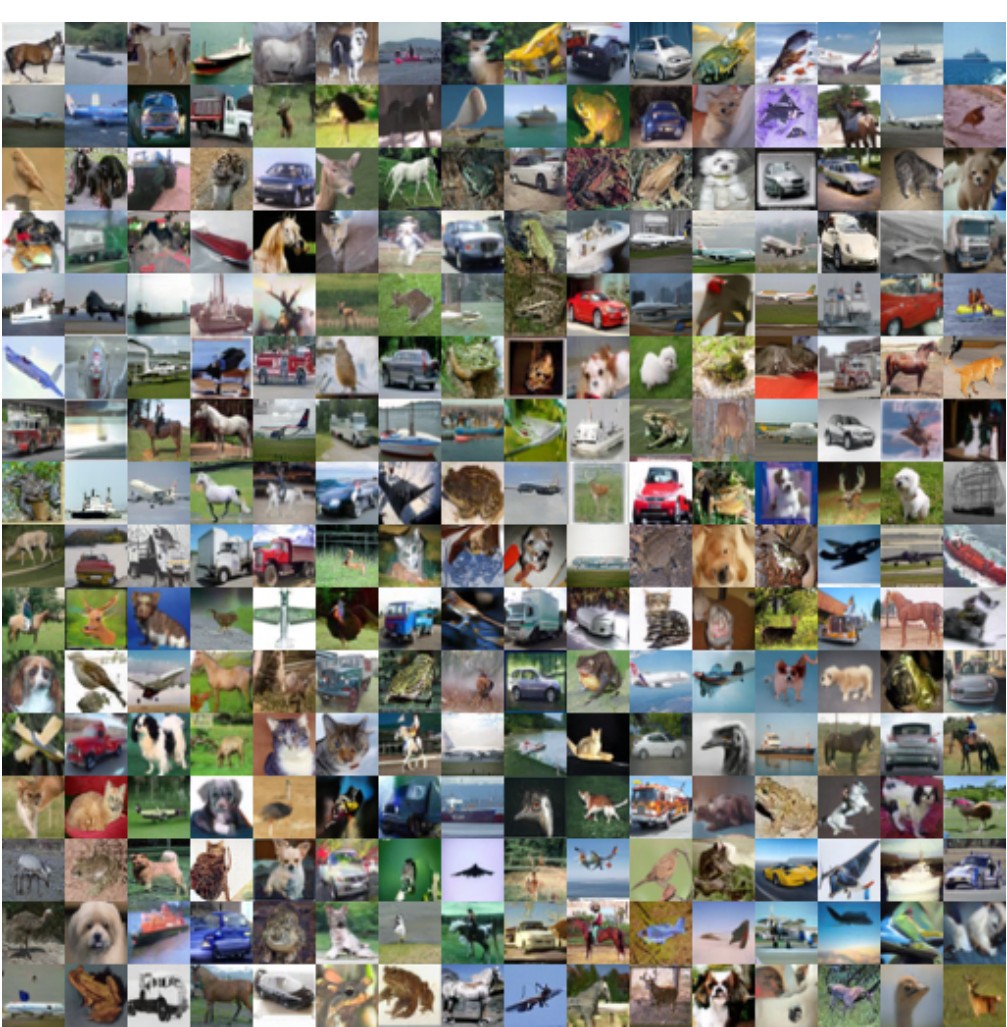

Figure 11: 4-step samples from unconditional SCT trained on CIFAR-10.

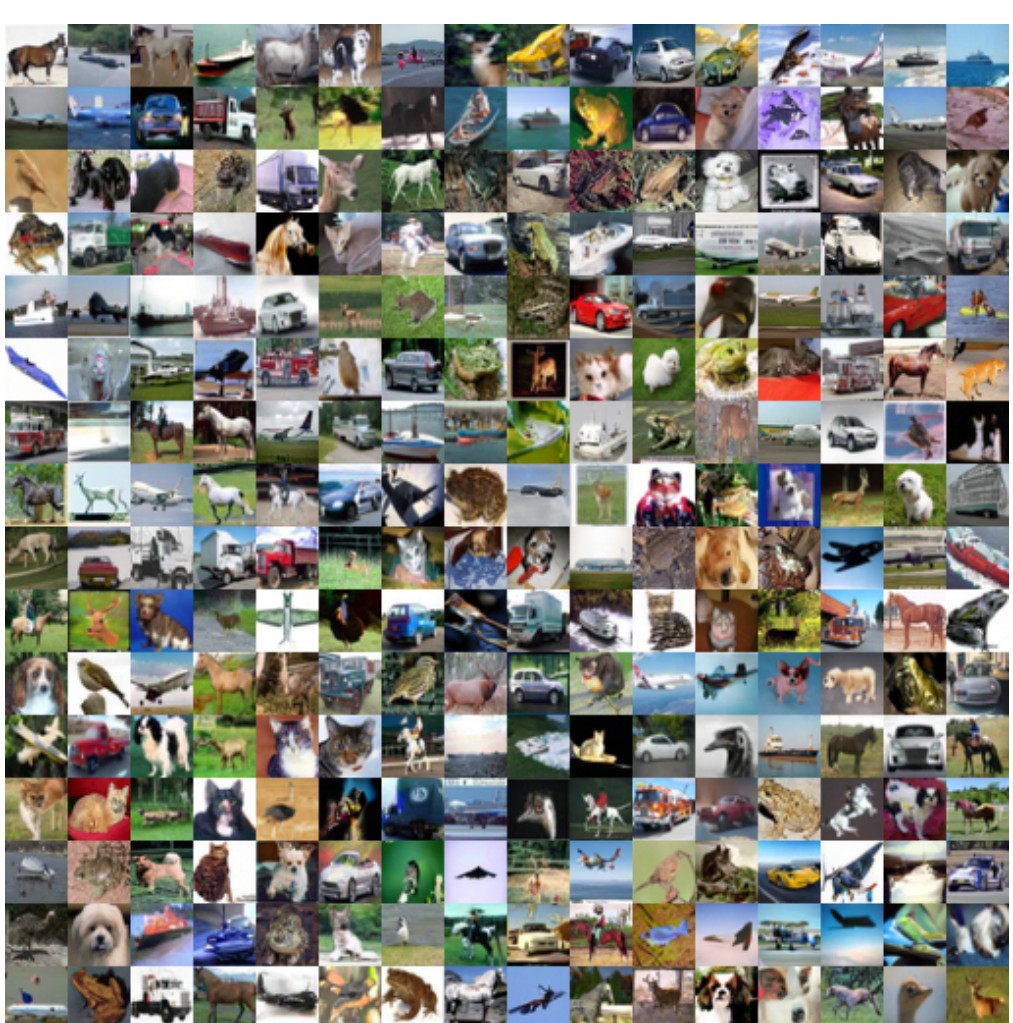

Figure 12: 8-step samples from unconditional SCT trained on CIFAR-10.

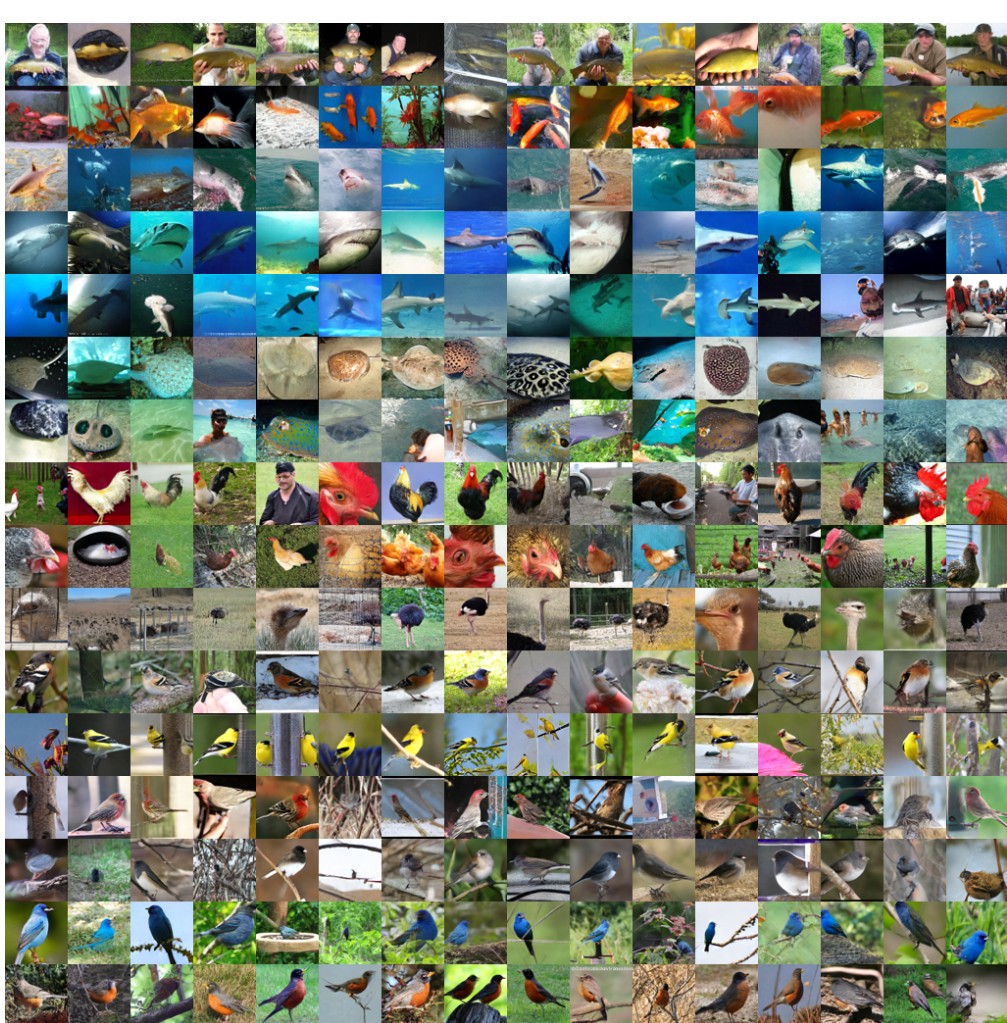

Figure 13: 1-step samples from class-conditional SCT trained on ImageNet-64 (FID 2.23). Each row corresponds to a different class.

1242
1243
1244
1245
1246
1247
1248
1249
1250
1251
1252
1253
1254
1255
1256
1257
1258
1259
1260
1261
1262
1263
1264
1265
1266
1267
1268
1269
1270
1271
1272
1273
1274
1275
1276
1277
1278
1279
1280
1281
1282
1283
1284
1285
1286
1287
1288
1289
1290
1291
1292
1293
1294
1295

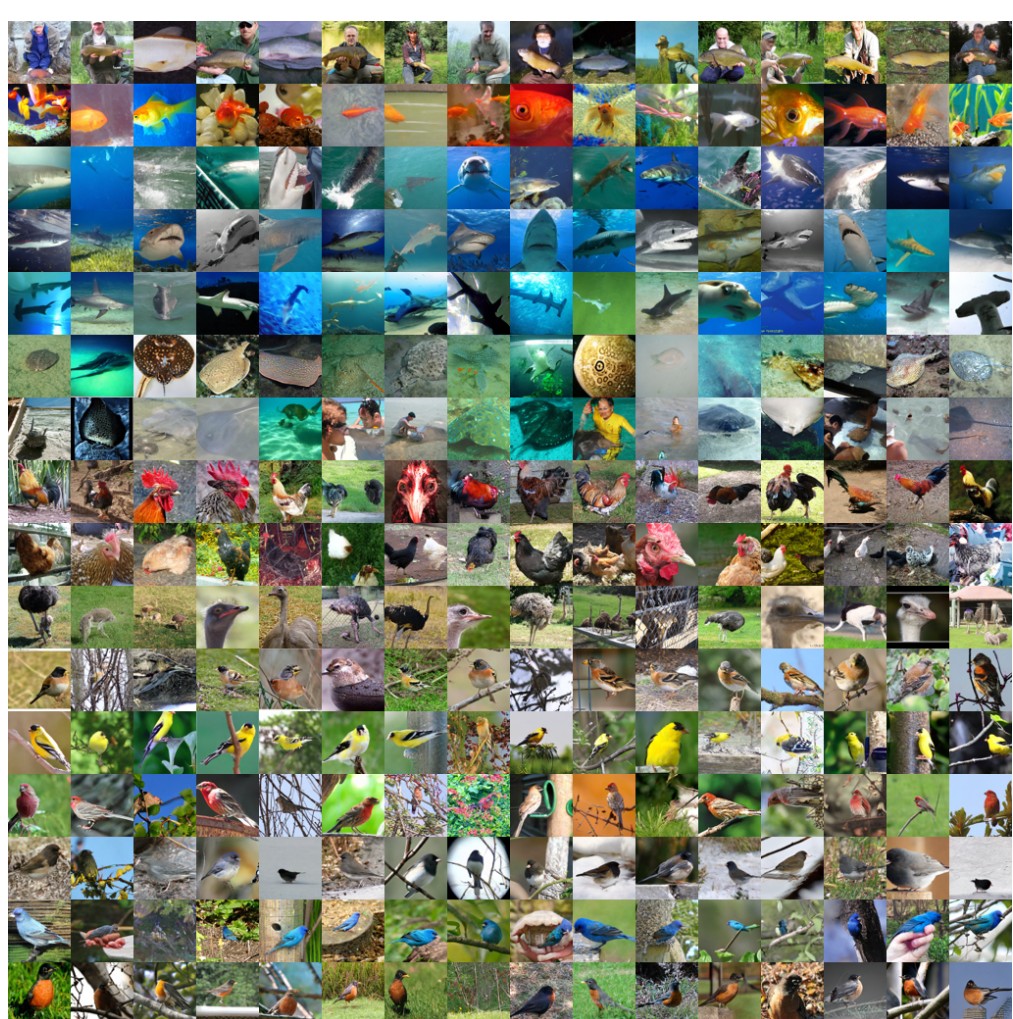

Figure 14: 2-step samples from class-conditional SCT trained on ImageNet-64 (FID 1.47). Each row corresponds to a different class.

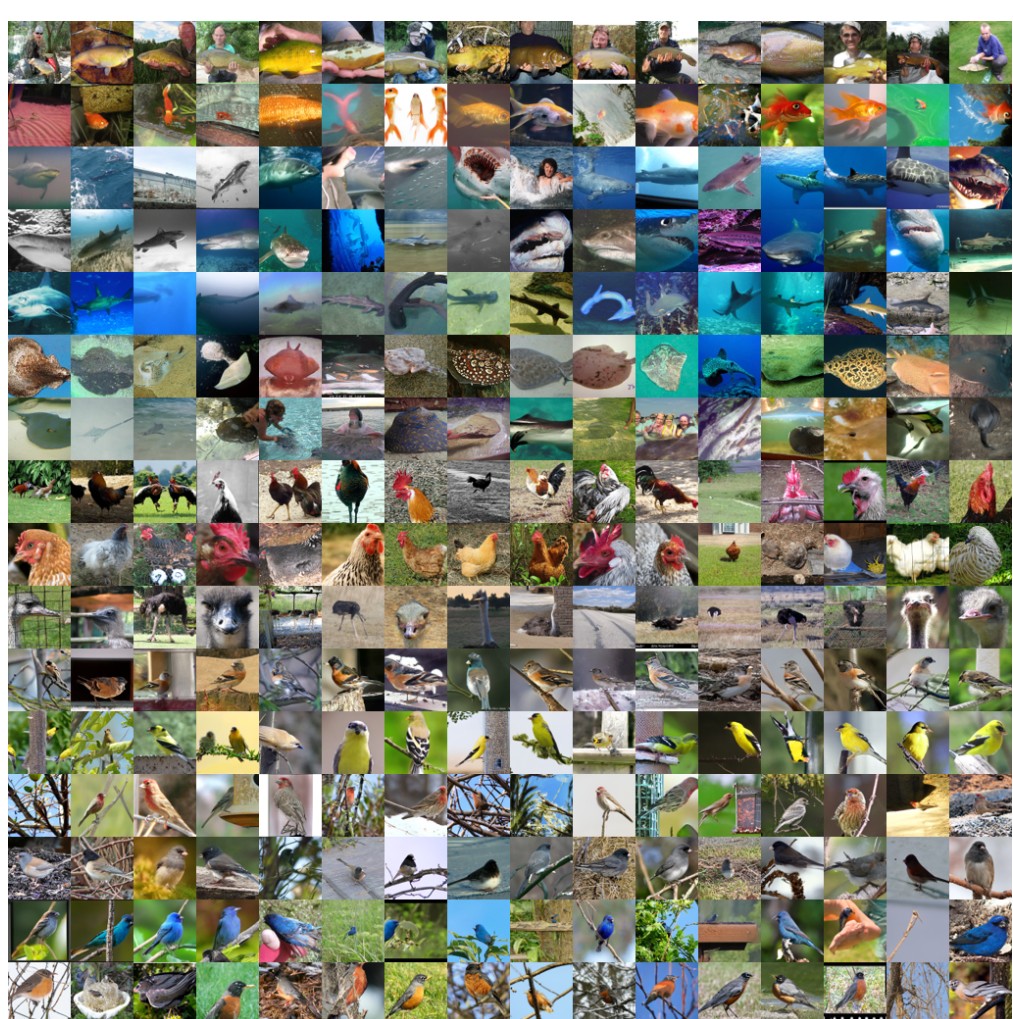

Figure 15: 4-step samples from class-conditional SCT trained on ImageNet-64 (FID 1.78). Each row corresponds to a different class.

