# OpenReview forum: "Stable Consistency Tuning: Understanding and Improving Consistency Models"
_ICLR.cc/2025/Conference — ICLR 2025 Conference Withdrawn Submission_

### Official Review · Reviewer_7pps · 2024-10-28

**Soundness:** 1
**Presentation:** 1
**Contribution:** 1
**Rating:** 1
**Confidence:** 4

**Summary:**

This paper frames the learning of consistency models as state value learning in a Markov decision process (MDP), deducing two main performance bottlenecks in consistency training: the variance of the score/noise estimator and the temporal discretization error. The authors then deploy several strategies to tackle these two bottlenecks, notably leveraging a better noise estimator and changing the temporal schedule. The resulting consistency model outperforms the state of the art on standard benchmarks.

**Strengths:**

The idea of **leveraging a noise estimator with lower variance**, taken from the literature of diffusion models, to improve consistency training is interesting. It tackles a well identified problem of consistency models and, based on the presented ablation study, seems to yield valuable results.

**Weaknesses:**

The paper presents too many important flaws, described below, to be considered for ICLR. Therefore, I strongly recommend rejecting the paper.

## Relevance of the MDP framework

The developed MDP framework is **disconnected from the rest of the contributions**. In the paper, the MDP view of consistency models provides no other insight related to the rest of the paper then than the identification of two main improvements directions -- variance reduction and alleviation of discretization errors. However, **this MDP view is not necessary** to identify these directions: it has already been clearly identified in prior work via more direct means.
- The discretization error is already identified in the original paper of Song et al. (2023, Theorem 2).
- Variance is already discussed by Song et al. (2023, Section 5), Song & Dhariwal (2023, Section 3.4) and Dou et al. (2023), for example. The variance problem can directly be identified as originating in the one-sample estimation of the score as already noted by Song et al. (2023, Section 5). It is the starting point of several research papers aiming to reduce this variance (Xu et al., 2023; Pooladian et al., 2023).

Dou et al. A Unified Framework for Consistency Generative Modeling. [ICLR 2024 submission](https://openreview.net/forum?id=Qfqb8ueIdy).\
Pooladian et al. Multisample flow matching: Straightening flows with minibatch couplings. ICML 2023.

## Limited novelty

The proposed solutions to the variance and discretization problems (Section 4) are **orthogonal and either already exist in the literature or are adaptions of existing techniques**. While combinations of existing methods can be acceptable, the contributions should be more developed, better motivated, and further supported by more experiments.

The most valuable novelty is the idea to use an already existing lower-variance estimator of the score/noise (Section 4.1) for consistency training. But this is insufficiently developed and tested. On the same topic, the paper highlights the adaption of this technique to conditional generation, but it is **straightforward**; furthermore, as the authors notice, it is not directly applicable to non-categorical conditioning.

## Unclear writing and unsupported claims

The writing of the paper is lacking in many aspects, even beyond the many typos (cf. next weakness) and leads several claims to be unsupported. **Section 3 is particularly unclear**, hurting the soundness of the MDP view.
- While Eq. (8) might be seen as a Bellman equation in a relevant setting, it is unclear to me why $h_\theta$ is a value function and $r$ a reward, especially as the latter is never optimized anywhere.
- The MDP was only developed for continuous-time consistency distillation at the beginning of the section, but is discussed in other settings (training and discretization) afterwards.
- Eq. (9), (11) and (12) lack context and development to be properly understood. They could be encompassed in small lemmas or propositions.
- The remark on the link with $n$-step temporal difference (line 198) is insufficiently explained.
- The introduction mentions performance bounds (lines 77 to 80), but they are nowhere to be found.

## Lacking finalization

Firstly, given the low significance of the contributions, **the experimental part of the paper deserves more focus**. Besides developing it, I would advice the authors to **release the source code** to ensure the paper's reproducibility.

Secondly, the paper suffers from too many mistakes or typos, making it hard to read, as shown in the following (possibly non-exhaustive) list.
- Mathematical problems:
  - $t$ and $r$ are not defined on line 81;
  - On line 114, the expectation should be over $x_0$ instead of $x_0 | x_t$;
  - On line 136, $t_\lambda$ should be the reverse function of $\lambda_t$, and $\mathbf{\epsilon}(\mathbf{x}\_{t\_\lambda, t})$ should be $\mathbf{\epsilon}(\mathbf{x}\_{t\_\lambda}, t\_\lambda)$;
  - In Eq. (7), $\alpha_s$ should be $\alpha_r$.
- Formatting:
  - the dataset is not specified for Figure 3's experiment;
  - I suspect that most of the figures are not properly readable in grayscale;
  - there should be no line between points in Figure 4's plot;
  - Figure 5 should be referenced in the corresponding paragraph ("Edge-skipping Multistep Sampling");
  - Tables 2 and 3's captions almost overlap;
  - lines 523 to 526 should be written using the correct font size;
  - it is not specified whether all CIFAR-10 experiments are conditional or unconditional;
  - a few references are duplicated (Song et al., 2020; Song et al., 2023).
- Typos/language:
  - on line 149: "weighg";
  - on line 154, "it is noting" is not correct English;
  - on line 188, "depent";
  - on line 196, "eough";
  - on line 202, "konwn";
  - on line 218, "gourd";
  - on lines 413 and 415, "LIPIPS";
  - on line 523, "a previous papers".

**Questions:**

I have no specific question -- cf. the above weaknesses for potential improvements of the paper. I recommend a "strong reject" and suggest the authors to submit a major revision to a later conference.

---

### Official Review · Reviewer_ZKwS · 2024-11-03

**Soundness:** 2
**Presentation:** 2
**Contribution:** 2
**Rating:** 3
**Confidence:** 5

**Summary:**

This paper shows the correspondence between the MDP and the consistency model, as they use the same temporal learning training objective. From the MDP view, they identify the performance bottlenecks in consistency tuning and consistency training, which are training variance and discretization error. For training variance, the paper adopts STF [1]. To reduce discretization error, the author uses a training scheduler from ECT [2] and a weighting technique from iCT [3]. Several sampling techniques, such as guided sampling and edge-skipping inference, are used in this paper.

[1]: Stable Target Field for Reduced Variance Score Estimation in Diffusion Models

[2]: ECT: Consistency Models Made Easy

[3]: Improved Techniques for Training Consistency Models

**Strengths:**

1. Viewing the consistency model as MDP is interesting.
2. Edge-skipping inference for MCM improves sampling FID.

**Weaknesses:**

1. The paper's novelty is limited. Since understanding CM as MDP is interesting, training variance and discretization error are already mentioned in previous work [1, 2].
3. Using STF for consistency training to tackle training variance seems straightforward. STF seems very hard to apply for text-to-image, limiting the application of this technique.
4. Techniques to reduce discretization error are also introduced from previous work [1,2].
5. Guiding consistency models from its bad version is from [3].
6. Lack of comparison between edge-skipping vs aDDIM from MCM [4].
7. The background of MCM should be provided since improved MCM is part of the method. The lack of proper background makes the paper hard to read.
8. Minor: there are typos in lines 217 and 218 (ground truth).

[1]: ECT: Consistency Models Made Easy

[2]: Improved Techniques for Training Consistency Models

[3]: Guiding a Diffusion Model with a Bad Version of Itself

[4]: Multistep Consistency Models

**Questions:**

My biggest concern is the novelty of this work. More experiments are needed to verify the effectiveness of STF for the conditional consistency model.
Since the author identifies training variance and discretization error as two main bottlenecks for consistency training/finetuning, the experiment only focuses on consistency tuning, which requires the pretrained diffusion model. There is no evidence that the proposed methods will work with consistent training. Furthermore, if the consistency model is initialized with a pretrained diffusion model, it already has low training variance. Therefore, it would be more interesting to see if STF works with consistency training.

---

### Official Review · Reviewer_bbvA · 2024-11-04

**Soundness:** 2
**Presentation:** 2
**Contribution:** 2
**Rating:** 3
**Confidence:** 5

**Summary:**

The paper views Consistency Models (CMs) as a Markov Decision Process (MDP) with learning objective as Temporal Difference (TD) comes from Reinforcement Learning (RL). Proposed method incorporates several techniques: variance-reduced training target (from STF[1]), smoothen timestep choices to reduce discretization errors (from ECT[2]), splitting time axis into multiple segments during training (from MCM[3], PCM[4]), guiding mechanism from [5], and edge-skipping inference for multistep model. In overall, the results have quite significant improvement in 1-step and 2-step evaluation.

[1] SFT: Stable target field for reduced variance score estimation in diffusion models

[2] ECT: Consistency Models Made Easy

[3] MCM: Multistep consistency models

[4] PCM: Phased Consistency Model

[5] Guiding a Diffusion Model with a Bad Version of Itself

**Strengths:**

The proposed Edge-skipping mechanism (4.4) in inference for multistep CM explained through L-Lipschitz continuity of neuron networks sounds interesting.

**Weaknesses:**

While the paper shows improvements in results, it seems to lack novelty. As mentioned above, most factors incorporated into the final method are drawn from relevant works, with little or no modification.

1. Reducing the training variance (4.1) basically comes from SFT[1]

2. Reducing the discretization error (4.2) has already been presented in ECT[2]

3. Guiding consistency models with a bad version of itself (4.4) from [3] is applied with modification in choice of the sub-optimal version of network.

Regarding the experiments:

4. Edge-skipping inference (4.4) is an interesting technique to enhance sample quality in multistep CMs, comparison with previous works (MCM[4], PCM[5]) also conducted in this settings is required to demonstrate its advantages and potential.

Furthermore:

5. Consistency tuning initializes the model from a pretrained Diffusion Model (DM), providing it with a good starting point. In contrast, for consistency training, a pretrained DM is not available, which may lead to greater training variance. Experiments demonstrating the advantage of variance reduction in consistency training could enhance the paper.

[1] SFT: Stable target field for reduced variance score estimation in diffusion models

[2] ECT: Consistency Models Made Easy

[3] Guiding a Diffusion Model with a Bad Version of Itself

[4] MCM: Multistep consistency models

[5] PCM: Phased Consistency Model

**Questions:**

I have some questions to the authors:

1. SCT has faster convergence speed and better performance than ECT. Can you show the additional computation cost of SCT compared to ECT, for example in GFLOPs, since SCT requires the conditional epsilon estimation (Equation (14))?

2. To clarify: Figure 6 illustrates the effective of CFG in three difference choices of the sub-optimal version of EMA weights (at 25%, 50%, and 75% of the total iterations). Does it mean that the guiding technique is only applied starting from one of these pivot iterations?

---

### Official Review · Reviewer_F1or · 2024-11-05

**Soundness:** 3
**Presentation:** 3
**Contribution:** 2
**Rating:** 5
**Confidence:** 4

**Summary:**

This paper analyzes the training bottlenecks of consistency models and proposes several improvements for better training of consistency models which are based on previous works of diffusion models, including a bias-variance trade-off training objective for consistency training; a smooth weighting function and reduced schedule of time spacing; a multistep version with edge-skipping and a better guidance techniques. Though all of these methods are not novel, the combination shows promising empirical results on CIFAR10 and ImageNet64.

**Strengths:**

- The paper explores several techniques in the recent progress of diffusion models and improves the empirical performance of consistency models.
- The methods are technically correct and easy to follow.
- The experiments are solid.

**Weaknesses:**

Though this paper proposes several techniques, overall it is kind of incremental and strongly increases the complexity of the algorithm. It is hard to tell which part is important and which part is not; and it is hard to tell which part plays a crucial role for the "stable" claim.

- The analysis framework and the final conclusions in section 3 are well-known in the community, and the training variance and discretization errors are not new understanding aspects. The proposed MDP framework seems to be unrelated to the contents in section 4.
- The proposed methods in section 4 are based on previous techniques in diffusion models, which all bring small improvements and cannot fundamentally "stabilize" the training of consistency models.

**Questions:**

- What's the relationship between the MDP in section 3 and the methods in section 4?
- What's the most significant part for stable training?

---

### Note · Authors · 2024-11-20

**Comment:**

We sincerely thank the reviewers for their feedback and valuable suggestions.

We are encouraged by the recognition of our **solid performance** and **novel interpretation of consistency models from reinforcement learning**. However, we acknowledge the reviewers’ constructive points.

To ensure that our contributions are as robust and impactful as possible, we have decided to withdraw our submission from ICLR 2025. This will allow us to:

- Strengthen the theoretical and experimental alignment of the MDP framework with our proposed methods to provide deeper insights and broader applicability.
- Expand our experimental validation to include more diverse benchmarks and settings, improving the generalizability of our results.
- Refine the manuscript’s clarity, enriching more experimental details, addressing the identified presentation and typographical issues to enhance readability and precision.
- Share our code to ensure reproducibility and foster further research in this direction.

We deeply value the feedback. Thank you for your time and constructive input.

**Withdrawal Confirmation:**

I have read and agree with the venue's withdrawal policy on behalf of myself and my co-authors.